# Targeted Hyperparameter Optimization with Lexicographic Preferences Over Multiple Objectives

**Shaokun Zhang**[1]**, Feiran Jia**[1]**, Chi Wang**[2]**, Qingyun Wu**[1]
[1] Pennsylvania State University, State College, PA, USA
[2] Microsoft Research, Redmond, Washington, USA
`{shaokun.zhang, feiran.jia@psu.edu, qingyun.wu}@psu.edu,`
`wang.chi@microsoft.com`

## Abstract

Motivated by various practical applications, we propose a novel and general formulation of targeted multi-objective hyperparameter optimization. Our formulation allows a clear specification of an automatable optimization goal using lexicographic preference over multiple objectives. We then propose a randomized directed search method named LexiFlow to solve this problem. We demonstrate the strong empirical performance of the proposed algorithm in multiple hyperparameter optimization tasks.

## 1 Introduction

Hyperparameter optimization (HPO) of machine learning models, as a core component of AutoML, is a process of finding a good choice of hyperparameter configuration that optimizes the model "performance". In the context of practical ML systems, there are typically more than one metrics to evaluate the model "performance" on which one desires to optimize. For instance, latency (He et al., 2018), fairness (Brookhouse & Freitas, 2022), and explainability (Gonzalez et al., 2021) are important complementary metrics of interest in addition to prediction accuracy in many application scenarios. Typical multi-objective HPO (MO-HPO) approaches (Knowles, 2006; Daulton et al., 2020) seek to find wide-spread Pareto frontiers for users to choose from. This type of method can only establish a partial ordering of the configurations. The final choice on which Pareto frontier to use is typically done manually and is opaque to the optimization algorithm. We call such optimization "untargeted". An automated approach is desirable, especially in repetitive tuning scenarios such as *continuous integration and delivery* (CI/CD) of machine learning models or MLOps in general (Garg et al., 2021; Mäkinen et al., 2021; Symeonidis et al., 2022). This automation is possible if the criteria for selecting the final choice is specified explicitly. In this scenario, untargeted HPO can be inefficient as the optimization algorithm may waste resources on finding Pareto frontiers that are far from the desired final choice, i.e., the target.

In this work, we consider a targeted HPO scenario: practitioners have a priority order over the objectives, which enables a total ordering of all the configurations. We formalize a general notion of priority order rigorously as a *lexicographic preference* (Fishburn, 1975) over multiple objectives in an HPO task. It allows users to specify a clear optimization target across multiple objectives before the optimization starts and removes the need for manual post hoc selection. Such a priority structure is found in HPO tasks from various application domains. For example, in many bioinformatics applications, besides the primary objective of finding model hyperparameter configurations with low prediction error, minimizing feature numbers via a feature selection step is found to be helpful in avoiding overfitting and discovering relevant features for domain experts and thus is suggested to be used as an auxiliary objective in HPO (Bommert et al., 2017; Gonzalez et al., 2021). When both objectives are included, the auxiliary objective is considered less important than the minimization of the prediction error, which naturally forms a lexicographic structure.

Despite its appealing practical importance, we find this type of targeted HPO problem remarkably under-explored. In this work, we first provide a rigorous problem formulation for the targeted HPO

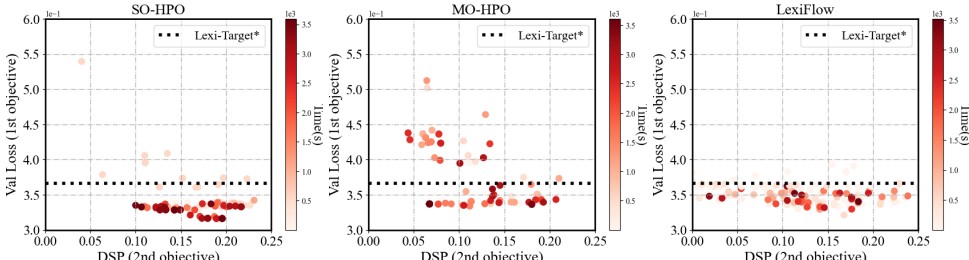

Figure 1: Results in an XGBoost tuning task to find accurate and fair models, in which validation loss is specified as an objective of a higher priority, and DSP (fairness-related objective) of a lower priority. Lexi-Target$^* = l^* + 0.05$, in which $l^*$ is the optimal loss value which is unknown before the optimization starts and $0.05$ is the user-specified tolerance onloss degradation. The circles represent proposed configurations from different methods in the objective space (darker color indicates later iteration). Both objectives are smaller the better. SO-HPO easily achieves the target loss value but performs poorly regarding fairness. MO-HPO is able to achieve a better performance in terms of fairness compared to SO-HPO. However, it also wastes resources on finding points outside the desired loss target as it seeks a wide spread of the Pareto front. Compared to MO-HPO, a larger fraction of the configurations in LexiFlow are within the loss target. This allows LexiFlow to try more configurations within the desired loss range and achieves better fairness performance.

task with lexicographic preference over multiple objectives. This formulation provides a general and flexible way for the users to specify customized targets expressed via a priority order on the objectives and a list of optional goals and tolerances on the objectives. Based on the problem formulation, we propose an algorithm named *LexiFlow* as a general solution. Specifically, LexiFlow conducts the optimization by leveraging pairwise comparisons between hyperparameter configurations in a randomized direct search framework. The pairwise comparisons are supported by a suite of targeted lexicographic relations, which allow us to navigate toward the more promising region of the search space considering the lexicographic structure in the objective space. By doing so, the algorithm is able to efficiently optimize the objectives with a strong any-time performance.

We perform extensive empirical evaluation on four different machine learning model tuning tasks, including a tuning task to find accurate and fast/small neural network models, a tuning task to find accurate and fair Xgboost models, a tuning task on random forest combined with feature selection for gene expression prediction, and a tuning task to mitigate overfitting. Our method has promising performance on all the evaluated tasks. The good empirical performance verified the unique advantages of our proposed algorithm. We demonstrate different performance patterns of methods including our method LexiFlow, a single objective method (SO-HPO), and a multiple objective method (MO-HPO) in Figure 1.

## 1.1 RELATED WORK

There are a number of works trying to address MO-HPO tasks on machine learning models, including evolutionary algorithms (Deb et al., 2002; Srinivas & Deb, 1994; Zhang & Li, 2007; Binder et al., 2020), Bayesian optimization (Knowles, 2006; Daulton et al., 2020; Emmerich & Klinkenberg, 2008; Hernández-Lobato et al., 2016) and multi-fidelity methods (Schmucker et al., 2020; 2021). All the aforementioned methods treat all the objectives equally important and seek an approximation of the Pareto front in the objectives space. Some recent work proposes to incorporate different types of user preferences into MO-HPO. Paria et al. (2020) allow users to encode their preferences as a prior on a weight vector to scalarize multiple objectives. The prior will induce a posterior distribution over the set of Pareto optimal values. Setting a proper prior in practice is non-trivial as the relation between the prior and posterior is difficult, if not impossible, to derive for an average practitioner. Abdolshah et al. (2019) regard preferences as the stability of objectives, using a constrained Bayesian optimization method. An earlier multi-objective optimization method (Zitzler et al., 2008) incorporates preference information into the multiple objective evolutionary frameworks by defining relations between different populations. However, this preference information is defined upon different populations rather than individual configurations.

## 2 PROBLEM FORMULATION

In this section, we first provide a rigorous formulation of the targeted HPO problem with lexicographic preferences over multiple objectives together with a motivating example. We then analyze the challenges in approaching the problem.

### 2.1 PROBLEM FORMULATION

Throughout the paper, we use $[K]$ to denote the integers from 1 to $K$. The studied problem needs the following inputs from the user,

- A $d$-dimensional hyperparameter search space $X$. Any hyperparameter configuration $\mathbf{x} \in X$ is considered a feasible configuration.
- A list of $K$ ($K > 1$) objective functions regarding $\mathbf{x} \in X$, denoted by $F(\mathbf{x}) = [f^1(\mathbf{x}), f^2(\mathbf{x}), ..., f^K(\mathbf{x})]$. Specifically, each $f^k$ ($\forall k \in [K]$) is a real-valued objective function of $\mathbf{x} \in X$, and the superscript $k$ in $f^k$ indicates the priority order (the smaller the index, the high the priority) of the objective. Without loss of generality, we assume minimization problems regarding all the objectives here and throughout this paper.
- A $K$-dimensional tolerance vector on the objectives denoted by $T = [\tau^1, \tau^2, ..., \tau^K]$. $\forall k \in [K]$, each $\tau^k$ is a non-negative number representing an optimality tolerance on the $k$-th objective. It can be intuitively considered as the amount of performance degradation the user is willing to compromise in order to find choices with better performance on the objectives of lower priorities.
- $C = [c^1, c^2, ..., c^K]$ a $K$-dimensional goal vector, in which $c^k \in R$ denotes a goal value on the $k$-th objective desired by the user.

With the inputs $F, T, C$ specified above, $\forall k \in [K]$, we can define a *lexi-target value* $z_*^k$. The lexi-target value $z_*^k$ of the $k$-th objective is defined recursively based on the lexi-target values of the more important objectives, i.e., the first $(k-1)$ objectives. We define $X_*^0 = X$ and for $k \in [K]$,

$$z_*^k := \max\{c^k, f_*^k + \tau^k\}, f_*^k := \inf_{\mathbf{x} \in X_*^{k-1}} f^k(\mathbf{x}), X_*^k := \{\mathbf{x} \in X_*^{k-1} | f^k(\mathbf{x}) \leq z_*^k\} \tag{1}$$

where $X_*^1 \supseteq X_*^2 \supseteq ... \supseteq X_*^K$ is a series of nested sets dubbed *lexi-frontiers* considering the first $1, 2, ..., K$ objectives respectively; and $f_*^k$ is the inferior limit of lexi-frontiers in $X_*^k$ on the $k$-th objective. Note that the cardinality of $\{F(\mathbf{x})\}_{\mathbf{x} \in X_*^K}$ can be larger than one. To compare these $F(\mathbf{x})$ vectors, we define lexicographic relations as follows:

$$F(\mathbf{x}') =_l F(\mathbf{x}) \Leftrightarrow f^k(\mathbf{x}') = f^k(\mathbf{x}) \ \forall k \in [K] \tag{2}$$

$$F(\mathbf{x}') \prec_l F(\mathbf{x}) \Leftrightarrow \exists k \in [K] : f^k(\mathbf{x}') < f^k(\mathbf{x}) \wedge (\forall k' < k, f^{k'}(\mathbf{x}') = f^{k'}(\mathbf{x})) \tag{3}$$

$$F(\mathbf{x}') \preceq_l F(\mathbf{x}) \Leftrightarrow F(\mathbf{x}') \prec_l F(\mathbf{x}) \vee F(\mathbf{x}') =_l F(\mathbf{x}) \tag{4}$$

Our aim is to find a *lexi-optimal* configuration, i.e., any element in $X^* = \{\mathbf{x} \in X_*^K | \forall \mathbf{x}' \neq \mathbf{x}, F(\mathbf{x}) \preceq_l F(\mathbf{x}')\}$, using minimal cost. We call this problem a *Lexi-HPO* problem in short.

**Remark 1** (Lexi-optimal vs Pareto frontiers)**.** The lexi-optimal solutions in $X^*$ are always a subset of the Pareto frontiers in the $K$-dimensional objective space. The relative location of $X_*^K$ in the Pareto frontiers depends on the lexicographic preferences and the relaxations imposed via tolerance and/or goals. In this way, the users could specify their own 'target' of interest in the Pareto frontiers in a straightforward way before the optimization starts.

**Remark 2** (On tolerances and goals)**.** Note that the two inputs tolerances $T$ and goals $C$ are both **optional**. They are introduced to help impose customized needs on the targeted optimum in flexible ways. When there is no tolerance and no particular goal on a particular objective $k$, one can just set $\tau^k = 0$ and $c^k = -\inf$ without affecting the nature of the problem.

To facilitate understanding of the Lexi-HPO problem and better motivate the studied scenario, we include a concrete example for gene expression prediction from the bio-informatics domain below.

*Example: In the HPO task for gene expression prediction from (Bommert et al., 2017), three minimization objectives are considered during the HPO process, including the model's prediction*

*error, the resulting feature numbers after feature selection, and feature selection instability. The three objectives are in descending priority order. Despite the priority order, when trying to find the optimal choice, it is okay to consider the configurations whose loss does not exceed $l^* + 0.05$, in which $l^*$ denotes the best loss one can achieve by any feasible hyperparameter configuration. In addition, a feature number of $500$ is considered good enough for the second objective. Say in this example $l^* = 0.1$ and $\mathbf{x}_A, \mathbf{x}_B, \mathbf{x}_C, \mathbf{x}_D$ are whole Pareto frontiers with performance $F(\mathbf{x}_A) = [0.2, 100, 0.1]$, $F(\mathbf{x}_B) = [0.1, 600, 0.2]$, $F(\mathbf{x}_C) = [0.13, 500, 0.2]$, and $F(\mathbf{x}_D) = [0.1, 300, 0.5]$.*

This example can be formulated as a Lexi-HPO problem, where the objectives are $F = [\text{loss}, \text{feature number}, \text{instability}]$, tolerances are $T = [0.05, 0, 0]$, and goals are $C = [\inf, 500, -\inf]$. Under the Lexi-HPO problem, we have the following conclusions: (1) $\mathbf{x}_C$ is the optimal choice among the whole frontiers; (2) Combining the first conclusion with Remark 1, we can also conclude that $\mathbf{x}_C$ is also the lexi-optimal choice in the whole space. This example also shows that the tolerances $T$ and goals $C$ are, in general, intuitive to understand and easy to set by practitioners because they are directly about each single objective value.

## 2.2 CHALLENGES

We want to first remind the readers that in the context of HPO of ML models, the objectives are mostly *black-box*, and it is also expensive to get function evaluations because it involves training and validating an ML model. The black-box nature of this problem makes methods that require analytic forms of the objectives, e.g., gradient-based methods (Gong et al., 2021) inapplicable. We also urge methods that are efficient and cost-frugal because of the expensive function evaluations and practical resource limitations. One potential solution for the Lexi-HPO problem is to use MO-HPO approaches to find Pareto frontiers and add a post-processing step to narrow them down to the targeted optimal points. This type of method can be inefficient because they strive to find a wide spread of Pareto frontiers while the users are actually only interested in a subset of them in a particular region. Another potential solution is to perform single objective HPO or constrained single objective HPO regarding each objective sequentially following the preference order. More specifically, when optimizing the $k$-th objective, one can impose constraints of the form $f^i(\mathbf{x}) \leq z_*^i, \forall i \in [k-1]$ and estimate $z_*^k$ according to the result. However, one tricky problem in this approach is how to allocate limited resources to the optimization of different objectives. The resource allocation directly affects how reliable the constraints are. For example, if we allocate insufficient resources to the first objective, we will not reach the optimal value on the first objective which loosens the constraints imposed on the first objective when trying to optimize objectives of lower priorities. If we allocate excessive resources to the first objective, the optimization of other objectives can be hindered due to a lack of resources. We include this approach as a baseline in the experiment section.

## 3 ALGORITHM

We propose an algorithm named LexiFlow to solve the Lexi-HPO problem. We adopt a randomized direct search framework that could direct the search to the targeted optimum based on lexicographic comparisons over pairs of configurations. During the iterative optimization process, the algorithm automatically adjusts its focus on the different objectives based on their priorities and room for improvement. For example, when there is no room for improvement on the first objective in a particular iteration, the algorithm will direct the search toward the region where the second objective could improve without hurting the optimality of the first objective. Since such adjustment is made at every iteration, the algorithm keeps the chance to improve every objective when the optimality has not been reached. By doing so, the algorithm is able to adaptively allocate optimization resources over multiple objectives in a flexible order while respecting the priority. This strategy can help the algorithm achieve a good any-time performance.

LexiFlow is presented in Alg. 1. It takes as inputs the objectives $F$, a goal vector $C$ (optionally), and a tolerance vector $T$ (optionally). When $C$ and $T$ are not provided, we just set $c^k$ to $-\infty$, and $\tau^k$ to $0, \forall k \in [K]$. After an initialization step, which sets an initial hyperparameter configuration $\mathbf{x}_0$ and an initial stepsize $\delta_{init}$, the algorithm proceeds as follows: At each iteration $i$, LexiFlow maintains an 'incumbent' point, denoted by $\mathbf{x}_i$. The 'incumbent' point is the point in whose neighboring area the algorithm samples randomized directions to get new points to try next. More specifically, a new direction $\mathbf{u}$ is sampled uniformly at random from a unit sphere (Line 4 in Alg. 1), which leads to

two newly proposed points $\mathbf{x}_i \pm \delta\mathbf{u}$ (considering both the sampled direction and its opposite direction). After evaluating the newly proposed point(s), i.e., obtaining $F(\mathbf{x}_i + \delta\mathbf{u})$ and potentially also $F(\mathbf{x}_i - \delta\mathbf{u})$, it decides whether to update the incumbent with one of the newly proposed points, i.e., $\mathbf{x}_i + \delta\mathbf{u}$ or $\mathbf{x}_i - \delta\mathbf{u}$. In LexiFlow, this decision is made via a carefully designed `Update` function. Following the same spirit of an existing randomized direct search-based single-objective HPO method (Wu et al., 2021), LexiFlow includes the restart and dynamic stepsize techniques to free the algorithm from local optimum and manual configuration of stepsize (line 9-11 of Alg. 1). We also adopt the same setting of the initial stepsize $\delta_{init}$ and stepsize lower bound $\delta_{lower}$ with the aforementioned work. The algorithm terminates if it runs out of resource budget and outputs a lexi-optimal configuration $\mathbf{x}^*$, which is maintained in a way such that it is lexi-optimal among all the evaluated configurations. Following the general principles of direct search (Kolda et al., 2003), the incumbent point shall be maintained in a way such that it is the best or at least a very promising point among the historically evaluated points (within a local search thread without considering restart). This general principle, together with an iterative random sampling of new points around the incumbent, could gradually lead the search to more promising region (Kolda et al., 2003; Wu et al., 2021). The algorithm makes a decision on whether to update the incumbent with a newly proposed point via the `Update` procedure attached at the end of Alg. 1 (line 12-18).

---

**Algorithm 1:** LexiFlow

**Input:** Objectives $F(\cdot)$, goals $C$ (optional) and tolerances $T$ (optional).

1. **Initialization:** Initial configuration $\mathbf{x}_0$, $i' = r = s = 0$, $\delta = \delta_{init}$;
2. Obtain $F(\mathbf{x}_0)$, and $\mathbf{x}^* \leftarrow \mathbf{x}_0$, $\mathcal{H} \leftarrow \{\mathbf{x}_0\}$, $Z_{\mathcal{H}} \leftarrow F(\mathbf{x}_0)$
3. **while** $i = 0, 1, ...$ **do**
4.      Sample $\mathbf{u}$ uniformly from unit sphere $\mathbb{S}$
5.      **if** `Update`$(F(\mathbf{x}_i + \delta\mathbf{u}), F(\mathbf{x}_i), Z_{\mathcal{H}})$ **then** $\mathbf{x}_{i+1} \leftarrow \mathbf{x}_i + \delta\mathbf{u}$, $i' \leftarrow i$;
6.      **else if** `Update`$(F(\mathbf{x}_i - \delta\mathbf{u}), F(\mathbf{x}_i), Z_{\mathcal{H}})$ **then** $\mathbf{x}_{i+1} \leftarrow \mathbf{x}_i - \delta\mathbf{u}$, $i' \leftarrow i$;
7.      **else** $\mathbf{x}_{i+1} \leftarrow \mathbf{x}_i$, $s \leftarrow s + 1$;
8.      $\mathcal{H} \leftarrow \mathcal{H} \cup \{\mathbf{x}_{i+1}\}$, and update $Z_{\mathcal{H}}$ according to Eq. (8)
9.      **if** $s = 2^{d-1}$ **then** $s \leftarrow 0$, $\delta \leftarrow \delta\sqrt{(i'+1)/(i+1)}$;
10.      **if** $\delta < \delta_{lower}$ **then**
11.          $r \leftarrow r + 1$, $\mathbf{x}_{i+1} \leftarrow N(\mathbf{x}_0, I)$, $\delta \leftarrow \delta_{init} + r$ // Random Restart
12. **Procedure** `Update`$(F(\mathbf{x}'), F(\mathbf{x}), Z_{\mathcal{H}})$:
13.      **if** $F(\mathbf{x}') \prec_{(Z_{\mathcal{H}})} F(\mathbf{x})$ ***Or*** $\left(F(\mathbf{x}') =_{(Z_{\mathcal{H}})} F(\mathbf{x})$ *and* $F(\mathbf{x}') \prec_l F(\mathbf{x})\right)$ **then**
14.          **if** $F(\mathbf{x}') \prec_{(Z_{\mathcal{H}})} F(\mathbf{x}^*)$ ***Or*** $\left(F(\mathbf{x}') =_{(Z_{\mathcal{H}})} F(\mathbf{x}^*)$ *and* $F(\mathbf{x}') \prec_l F(\mathbf{x}^*)\right)$ **then**
15.              $\mathbf{x}^* \leftarrow \mathbf{x}'$ // Using $\mathbf{x}^*$ to keep track of the lexi-optimal solution
16.          **Return True**        // Accept $\mathbf{x}'$
17.      **else**
18.          **Return False**        // Discard $\mathbf{x}'$
19. **Output:** A lexi-optimal configuration $\mathbf{x}^*$

---

**Update Function.** Existing work on randomized direct search (Kolda et al., 2003; Wu et al., 2021), especially the one about cost-frugal HPO (Wu et al., 2021) indicates the following desiderata for doing cost-efficient randomized direct search: (1) Whenever a newly proposed point is better than the incumbent, it should be accepted as the new incumbent. This property can help achieve a good anytime performance of the algorithm and avoids getting stuck into local optimum while maintaining a good convergence rate. (2) The algorithm should not accept a point that is strictly worse than the incumbent. This property can help control the evaluation cost in each individual trial according to theoretical analysis of the cost of a randomized direct search method (Proposition 4 and 5 in (Wu et al., 2021)). Our `Update` function is designed following these desiderata.

One seemingly straightforward idea for the `Update` function is to directly make comparisons between the incumbent and the newly proposed points according to the vanilla lexicographic relations defined in Eq. (2), Eq. (3) and Eq. (4). However, one problem with vanilla lexicographic relations is that they cannot accommodate user-specified tolerances and goals on the objectives. To address this challenge, we introduce the *targeted lexicographic relations*, which are parameterized by a $K$-dimensional real-valued target vector $Z = [z^1, z^2, ..., z^K]$ and denoted by $=_{(Z)}, \prec_{(Z)}, \preceq_{(Z)}$,

$$F(\mathbf{x}') =_{(Z)} F(\mathbf{x}) \Leftrightarrow f^k(\mathbf{x}') = f^k(\mathbf{x}) \vee (f^k(\mathbf{x}') \leq z^k \wedge f^k(\mathbf{x}) \leq z^k) \ \forall k \in [K] \tag{5}$$

$$F(\mathbf{x}') \prec_{(Z)} F(\mathbf{x}) \Leftrightarrow \exists k \in [K]: f^k(\mathbf{x}') < f^k(\mathbf{x}) \wedge f^k(\mathbf{x}) > z^k \wedge F^{k-1}(\mathbf{x}) =_{(Z)} F^{k-1}(\mathbf{x}') \tag{6}$$

$$F(\mathbf{x}') \preceq_{(Z)} F(\mathbf{x}) \Leftrightarrow F(\mathbf{x}') \prec_{(Z)} F(\mathbf{x}) \vee F(\mathbf{x}') =_{(Z)} F(\mathbf{x}) \tag{7}$$

in which $F^{k-1}(\mathbf{x})$ denotes a vector with the first $k-1$ dimensions of $F(\mathbf{x})$, i.e., $F^{k-1}(\mathbf{x}) = [f^1(\mathbf{x}), ..., f^{k-1}(\mathbf{x})]$ (We define $F^0(\mathbf{x}) = 0, \forall \mathbf{x} \in X$). It is easy to verify that the relation $\preceq_{(Z)}$ is reflexive in $X$ and we prove in Lemma 1 (in Appendix A) that it is also transitive in $X$. With these properties, it can be verified that when $Z$ is set to the lexi-targets $Z_* = [z_*^1, z_*^2, ..., z_*^K]$, the targeted lexicographic relations can be used to find points that are in $X_*^K$, i.e., the lexi-frontiers, through exhaustive pairwise comparisons of configurations from the search space $X$.

Another caveat in this idea is that the comparisons depend on the knowledge of lexi-targets $z_*^k$ $\forall k \in [K]$, which are generally unknown during the optimization process. To address this difficulty, we propose to maintain an online approximate of the lexi-targets based on historical observations. Specifically, we maintain all historically evaluated points in $\mathcal{H}$ and introduce the following statistics of $\mathcal{H}$ based on Eq. (1): $X_{\mathcal{H}}^{(0)} = \mathcal{H}$ and $\forall k \in [K]$,

$$z_{\mathcal{H}}^k := \max\{c^k, f_{\mathcal{H}}^k + \tau^k\}, f_{\mathcal{H}}^k := \min_{\mathbf{x} \in X_{\mathcal{H}}^{k-1}} f^k(\mathbf{x}), X_{\mathcal{H}}^k := \{\mathbf{x} \in X_{\mathcal{H}}^{k-1} | f^k(\mathbf{x}) \le z_{\mathcal{H}}^k\} \quad (8)$$

Here $z_{\mathcal{H}}^k$, $f_{\mathcal{H}}^k$, and $X_{\mathcal{H}}^k$ can be considered online versions of $z_*^k$, $f_*^k$, and $X_*^k$ respectively, calculated based on historical observations in $\mathcal{H}$. With the approximated lexi-targets $Z_{\mathcal{H}} = [z_{\mathcal{H}}^1, z_2^{\mathcal{H}}, ..., z_{\mathcal{H}}^K]$, we decide whether to accept a newly proposed point $\mathbf{x}'$ comparing the the current incumbent $\mathbf{x}$ according to the `Update` procedure in Alg. 1. Specifically, we accept $\mathbf{x}'$ as the new incumbent in the following two cases: (1) When $F(\mathbf{x}') \prec_{(Z_{\mathcal{H}})} F(\mathbf{x})$ is true. This condition allows us to leverage the targeted lexicographic relation $\prec_{(Z_{\mathcal{H}})}$ to direct the search toward the approximated lexi-frontiers based on the approximated lexi-targets $Z_{\mathcal{H}}$. (2) When $F(\mathbf{x}') =_{(Z_{\mathcal{H}})} F(\mathbf{x})$ and $F(\mathbf{x}') \prec_l F(\mathbf{x})$. This condition allows us to move toward lexi-optimal points (also approximated) when the proposed point is already an approximated lexi-frontier. The update rules on the one hand respect the lexicographic relations considering the existence of lexi-targets and on the other hand follow the established desiderata. In Alg. 1 we use $\mathbf{x}^*$ to keep track of the lexi-optimal solution and update it whenever needed in a similar way in the `Update` procedure (line 13-15). Note that when there is no random start, $\mathbf{x}_*$ is simply the current incumbent, i.e., $\mathbf{x}_i$ at iteration. The extra update steps in line 13-15 are essential considering the potential random restarts.

## 4 EXPERIMENTS

We perform evaluations on multiple hyperparameter tuning tasks, which fall into the following two categories of use case scenarios: (1) To directly perform lexicographic optimization in cases where the users do have lexicographic preferences over multiple objectives. (2) To optimize a particular objective that is inaccessible during the optimization, by finding lexi-optimal solutions in terms of a list of proxy objectives. For the first type of use case, we include an efficient NN tuning task, an unfairness mitigation task, and a high-dimensional feature selection task in Section 4.1. For the second type of use case, we include a task about overfitting mitigation in Section 4.2. We tune three types of models including Xgboost, Random Forest and Neural Networks on different classification tasks. In all the experiments, if not otherwise specified, we show the results of the lexi-optimal solutions calculated based on the historical observations in each method at each particular concerned time point. All reported results are averaged over five runs with different random seeds.

### 4.1 HPO WITH LEXICOGRAPHIC PREFERENCES OVER MULTIPLE OBJECTIVES

Since there are no existing HPO methods that are directly applicable to the Lexi-HPO problem, we include three general HPO methods which could be extended to the Lexi-HPO problem: (1) A single objective HPO algorithm (SO-HPO), named CFO (Wu et al., 2021), which only optimizes the objective of the highest priority. The method CFO is selected as it is a strong SO-HPO method showing superior performance over other alternatives, e.g., BO-based methods, multi-fidelity-based methods as reported in (Wu et al., 2021). (2) A constrained optimization method (C-HPO) as described and constructed in Section 2.2. This method allocates even resources to different objectives. (3) A popular multiple objective HPO algorithm (MO-HPO) qEHVI (Daulton et al., 2020). We choose qEHVI in this category because it is a state-of-the-art MO-HPO method with ready-to-use open-source implementation. We use both CFO and its constrained optimization version implementation from the AutoML library FLAML (Wang et al., 2021), and use the implementation for qEHVI from the HPO library Optuna (Akiba et al., 2019). In all the experiments, when presenting the anytime

performance, we use a horizontal dashed line labeled **Lexi-Target**[*] to represent the lexi-target calculated based on user inputs and the configurations found by all the methods (unknown before the optimization) according to Eq. (8).

### 4.1.1 TUNING ACCURATE AND EFFICIENT NEURAL NETWORKS

Deploying neural networks in practical applications typically requires a balance between model accuracy and efficiency. Lexico-HPO can be a good way to find models achieving the desired balance. For example, as mentioned in (He et al., 2018), for AI applications such as Google Photos, model accuracy is more important than latency, and it is desirable to find smaller models under the condition of not sacrificing accuracy. There is a natural lexicographic preference over the two objectives of improving model accuracy and reducing latency. In this experiment, we evaluate our method for the task of tuning accurate and efficient neural networks. More specifically, following a similar search space setting with (Abdolshah et al., 2019; Hernández-Lobato et al., 2016), we tune neural network models on a subsampled Fashion MNIST dataset (Xiao et al., 2017) with two minimization objectives under lexicographic preference. The 1st objective is the error rate. The 2nd objective is efficiency-related objectives including FLOPs and parameter numbers in two different experiments respectively. For both experiments, we provide tolerance vector $T = [\tau^1 = 1/\sqrt{S}, \tau^2 = 0.0]$, where $S$ represents the number of validation data points since the error metric calculated from validation data can deviate from the true test error in the scale of $1/\sqrt{S}$. We do not set a specific goal vector for this task, as mentioned in (He et al., 2018), latency is typically not a hard constraint.

We perform tuning for 2 hours with each method and show the anytime performance of the methods regarding the two objectives in Figure 2. All four methods eventually achieve good performance in terms of the objective of the highest priority, i.e., the error rate: all of them are within the calculated lexi-target. LexiFlow, SO-HPO, and C-HPO achieve the target value earlier than MO-HPO because they put more priority on the first objective. In terms of the second objective, LexiFlow shows a significant performance improvement in both two experiments compared to all the baselines. MO-HPO finds models with lower FLOPs and lower parameter numbers than SO-HPO and C-HPO. But the FLOPs and parameter numbers in MO-HPO are still 2-4 times larger than that in LexiFlow when the error rate is within the tolerance range.

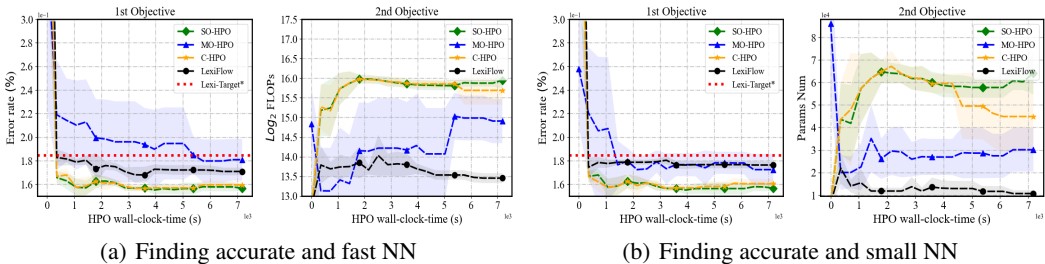

(a) Finding accurate and fast NN

(b) Finding accurate and small NN

Figure 2: Results from the neural networks tuning task on Fashion-MNIST. Each of the curves represents averaged result of a particular method from 5 runs with different random seeds. The shaded area corresponds to 95% confidence intervals.

### 4.1.2 TUNING ACCURATE AND FAIR XGBOOST

Due to increasing evidence on fairness-related harms (Angwin et al., 2016; Barocas & Selbst, 2016) in ML, there is an emerging need to build ML models that is not only accurate but also fair. One way to achieve this goal is to optimize the predictive performance of a model (1st objective) while minimizing the unfair bias (2nd objective) under a lexicographic preference in model tuning (Brookhouse & Freitas, 2022). Following the same setting with (Brookhouse & Freitas, 2022), we perform the Xgboost tuning task with two minimization objectives, including validation loss (1st objective) according to (Brookhouse & Freitas, 2022), and DSP (2nd objective), which an unfairness-related metric calculated based on statistical disparity (Perrone et al., 2021). We do the evaluation on datasets including Adult, Compas, and German, which are commonly used datasets for fairness-

related tasks (Perrone et al., 2021). We set a tolerance of 0.05 on the validation loss according to (Brookhouse & Freitas, 2022), and we do not provide the goal value .

We perform tuning for 1 hour with each method and show the final performance in Table 1. The results show that LexiFlow can indeed achieve the best performance considering the lexicographic preference over the objectives compared to baselines: LexiFlow has a similar performance with baselines considering optimality tolerance (0.05) in terms of the 1st objective and achieves a better performance in terms of the 2nd objective across all three datasets. Regarding the performance on the 2nd objective, i.e., the DSP value, 0.05 is a commonly used reference value for good DSP (Perrone et al., 2021). LexiFlow reaches 0.016 in Compas and 0.034 in German, where most of the other baselines got larger than 0.05.

Table 1: Results from tuning Xgboost on tasks that involve fairness considerations. All four methods are within the optimality tolerance range in 1st objective. In terms of the 2nd objective, LexiFlow achieves a significant performance improvement compared to baselines.

| Method | Compas | | German | | Adult | |
|---|---|---|---|---|---|---|
| | Val Loss | DSP | Val Loss | DSP | Val loss | DSP |
| SO-HPO | **0.344 (+0.004)** | 0.067 | **0.431 (+0.007)** | 0.064 | **0.368 (+0.003)** | 0.082 |
| MO-HPO | **0.340 (+0.000)** | 0.064 | **0.472 (+0.048)** | 0.078 | **0.408 (+0.043)** | 0.198 |
| C-HPO | **0.354 (+0.014)** | 0.029 | **0.424 (+0.000)** | 0.100 | **0.369 (+0.004)** | 0.088 |
| LexiFlow | **0.345 (+0.005)** | **0.016** | **0.434 (+0.010)** | **0.034** | **0.365 (+0.000)** | **0.060** |

### 4.1.3 FEATURE SELECTION IN BIOINFORMATICS

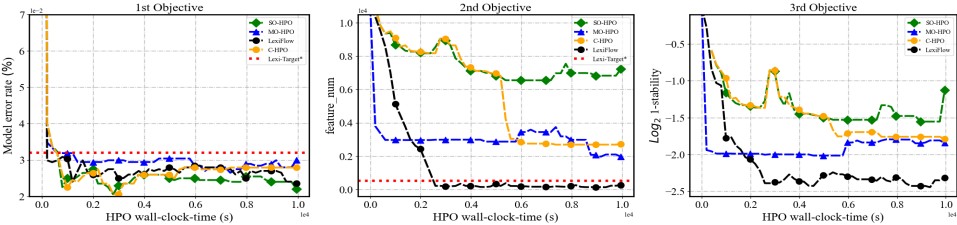

Figure 3: The results in tuning Xgboost on biological dataset AP_Colon_Kidney.

In many applications of ML in bioinformatics, the primary goal is to find a model with good predictive performance. For high-dimension biological data, due to a large number of features, it is also critical to integrate feature selection into the model fitting process, to make the selected features few and stable (Bommert et al., 2017), which could help provide insights for domain experts.

In this experiment, we verify LexiFlow in a feature selection task following a similar setting with (Bommert et al., 2017) which implicitly formulates the optimization objectives as lexicographic preference. Specifically, we perform the Xgboost tuning on biological dataset AP_colon_Kidney with three minimizing objectives including error rate, feature number, and the instability of the selected features (with descending priority order). We set targets $C = [c^1 = 0, c^2 = 500, c^3 = 0]$ and tolerances $T = [\tau^1 = 0.01, \tau^2 = 0.0, \tau^3 = 0.0]$ according to(Bommert et al., 2017) We use the filter feature selection method from Scikit-learn (Pedregosa et al., 2011) for feature selection.

We perform tuning for $10^4$ seconds and report the anytime performance regarding the three objectives in Figure 3. From the results, we observe that regarding the 1st objective, all four methods are within the optimality tolerance range. In terms of the 2nd objective, only LexiFlow reaches the goal in the end. Moreover, LexiFlow achieves significant performance improvement compared to baselines in the 3rd objective. We also include results from all five random seeds in Appendix B.

## 4.2 LEXIFLOW AS AN OPTIMIZATION TOOL TO MITIGATE OVERFITTING

Although low loss on the final test data is the objective of interest in machine learning, validation loss is commonly used as a proxy metric for test loss since the final test data is inaccessible in

Table 2: Test-time error rate (%) achieved by different methods on different datasets.

| Method | Gisette | Madelon | Ginal_prior | Ginal_agnostic | Bioresponse | Hiva_agnostic | Scene | Christine |
|---|---|---|---|---|---|---|---|---|
| SO-HPO w/o FS | $2.46 \pm 0.08$ | $17.1 \pm 0.7$ | $4.54 \pm 0.3$ | $5.00 \pm 0.2$ | $2.15 \pm 0.7$ | $2.41 \pm 0.2$ | $\mathbf{2.49 \pm 0.0}$ | $26.4 \pm 0.5$ |
| SO-HPO w/ FS | $2.30 \pm 0.2$ | $18.5 \pm 1.2$ | $5.01 \pm 0.4$ | $4.71 \pm 0.1$ | $\mathbf{2.14 \pm 0.7}$ | $2.39 \pm 0.1$ | $2.52 \pm 0.07$ | $26.5 \pm 0.9$ |
| LexiFlow w/ FS | $\mathbf{2.08 \pm 0.3}$ | $\mathbf{16.3 \pm 2.5}$ | $\mathbf{4.50 \pm 0.4}$ | $\mathbf{4.45 \pm 0.2}$ | $2.19 \pm 0.6$ | $\mathbf{2.36 \pm 0.1}$ | $\mathbf{2.49 \pm 0.0}$ | $\mathbf{26.1 \pm 1.0}$ |

the model building and tuning stage. It is well known that there may exist a gap between an ML model's performance on validation data and test data. Overfitting is a major factor that causes such a performance gap. Evidence from existing work show that optimizing machine learning models with fewer but key features could mitigate overfitting problems (Jović et al., 2015; Ying, 2019; Abdel-Aal, 2005). These findings inspire us to explore the possibility of incorporating the objective of minimizing feature numbers as an additional proxy objective in ML model tuning. Specifically, we consider the commonly used validation loss as the primary objective and feature number as a secondary proxy objective during HPO. We tune Random Forest (RF) with training and validation data and evaluate the best model's test loss on a reserved test dataset as the final performance metric.

More specifically, we include the following methods in comparison: (1) LexiFlow w/ FS, which is LexiFlow with two minimization objectives, including validation loss (1st), and feature number (2nd). (2) SO-HPO w/ FS, which is the single objective HPO algorithm CFO (Wu et al., 2021) with validation loss as its objective. (3) SO-HPO w/o FS, which is the single objective HPO algorithm CFO with validation loss as the objective and without a feature selection process. Both LexiFlow w/ FS and SO-HPO w/ FS include a feature selection step (indicated by w/ FS) using the same feature selection method as that in Section 4.1.3, in addition to RF model hyperparameter tuning. We include eight classification datasets, which are datasets from two previous feature selection studies (Gonzalez et al., 2021; Bommert et al., 2020) satisfying the following two conditions: feature number larger than 100 and available on Openml. We use 0.01 or 0.001 as the tolerance on validation loss according to empirical studies from (Gonzalez et al., 2021): for each dataset, if the mean validation loss from a default RF model (from sklearn) is larger than 0.1, 0.01 is used as the tolerance, otherwise 0.001. No tolerance or target is imposed on the 2nd objective.

We show the final test result from different methods in Table 2. LexiFlow w/ FS achieves the best performance on most of the datasets (7/8). We also investigate the number of features selected in each method and find that LexiFlow w/ FS achieves the smallest reserved feature ratio (30.1%) compared with SO-HPO w/o FS (100%) and SO-HPO w/ FS (66.8%). There are two important takeaways from this experiment, considering the nature of the compared methods and their performance: (1) The good test performance of SO-HPO w/o FS and LexiFlow w/ FS (over SO-HPO w/o FS) indicates that reducing feature number can indeed help mitigate overfitting and thus improve test performance, which is consistent with findings from existing work mentioned; (2) The dominating performance of LexiFlow w/ FS over SO-HPO w/ FS indicates that leveraging feature number as a secondary objective in addition to validation loss during model tuning is an effective way to further mitigate overfitting. Our method serves as an ideal optimization tool in this important endeavor.

## 5 CONCLUSION

In this paper, we propose an HPO algorithm named LexiFlow, which could easily incorporate users' lexicographic preferences across multiple objectives in HPO tasks. LexiFlow is simple and effective, showing a strong empirical performance over a wide spectrum of tuning tasks and application domains. LexiFlow is of good practical importance especially considering the ubiquitous existence of potentially conflicting objectives in modern machine learning systems, such as accuracy, latency, model size, robustness, and ethics-related objectives. We also made an interesting finding on overfitting mitigation: tuning certain proxy objectives with a lexicographic structure could help find models that are less likely to overfit. The implementation of our method is available in the open-source AutoML library FLAML[1].

---

[1]Link to the documentation page of LexiFlow in FLMAL: `https://microsoft.github.io/FLAML/docs/Use-Cases/Tune-User-Defined-Function#lexicographic-objectives`. code example demonstrating the use of LexiFlow to find accurate and fast neural networks: `https://microsoft.github.io/FLAML/docs/Examples/Tune-Lexicographic-objectives`.

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

## A  PROOF

**Lemma 1.** *The binary relation $\preceq_{(Z)}$ defined in Eq. (7) is transitive, i.e., $\forall \mathbf{x}, \mathbf{y}, \mathbf{w} \in X$, $F(\mathbf{x}) \preceq_{(Z)} F(\mathbf{y}) \wedge F(\mathbf{y}) \preceq_{(Z)} F(\mathbf{w}) \Rightarrow F(\mathbf{x}) \preceq_{(Z)} F(\mathbf{w})$.*

*Proof of Lemma 1.* According to Eq. (7),

$$
\begin{aligned}
&F(\mathbf{x}) \preceq_{(Z)} F(\mathbf{y}) \wedge F(\mathbf{y}) \preceq_{(Z)} F(\mathbf{w}) \Rightarrow \\
&\left(F(\mathbf{x}) \prec_{(Z)} F(\mathbf{y}) \vee F(\mathbf{x}) =_{(Z)} F(\mathbf{y})\right) \wedge \left(F(\mathbf{y}) \prec_{(Z)} F(\mathbf{w}) \vee F(\mathbf{y}) =_{(Z)} F(\mathbf{w})\right) \Rightarrow \\
&\underbrace{\left(F(\mathbf{x}) =_{(Z)} F(\mathbf{y}) \wedge F(\mathbf{y}) =_{(Z)} F(\mathbf{w})\right)}_{i} \vee \underbrace{\left(F(\mathbf{x}) \prec_{(Z)} F(\mathbf{y}) \wedge F(\mathbf{y}) \prec_{(Z)} F(\mathbf{w})\right)}_{ii} \\
&\vee \underbrace{\left(F(\mathbf{x}) =_{(Z)} F(\mathbf{y}) \wedge F(\mathbf{y}) \prec_{(Z)} F(\mathbf{w})\right)}_{iii} \vee \underbrace{\left(F(\mathbf{x}) \prec_{(Z)} F(\mathbf{y}) \wedge F(\mathbf{y}) =_{(Z)} F(\mathbf{w})\right)}_{iv}
\end{aligned}
\tag{9}
$$

**Statement I:** $F(\mathbf{x}) =_{(Z)} F(\mathbf{y}) \wedge F(\mathbf{y}) =_{(Z)} F(\mathbf{w}) \Rightarrow F(\mathbf{x}) =_{(Z)} F(\mathbf{w})$.

*Proof of Statement I.* From Eq. (5), we have,

$$
\begin{aligned}
&F(\mathbf{x}) =_{(Z)} F(\mathbf{y}) \wedge F(\mathbf{y}) =_{(Z)} F(\mathbf{w}) \Rightarrow \\
&\left(f^k(\mathbf{x}) = f^k(\mathbf{y}) \vee \left(f^k(\mathbf{x}) \le z^k \wedge f^k(\mathbf{y}) \le z^k\right), \forall k \in [K]\right) \\
&\wedge \left(f^k(\mathbf{y}) = f^k(\mathbf{w}) \vee \left(f^k(\mathbf{y}) \le z^k \wedge f^k(\mathbf{w}) \le z^k, \forall k \in [K]\right)\right) \\
&\Rightarrow \forall k \in [K] : \left(f^k(\mathbf{x}) = f^k(\mathbf{y}) = f^k(\mathbf{w})\right) \\
&\vee \left(\left(f^k(\mathbf{x}) = f^k(\mathbf{y})\right) \wedge \left(f^k(\mathbf{y}) \le z^k \wedge f^k(\mathbf{w}) \le z^k\right)\right) \\
&\vee \left(\left(f^k(\mathbf{x}) \le z^k \wedge f^k(\mathbf{y}) \le z^k\right) \wedge \left(f^k(\mathbf{y}) = f^k(\mathbf{w})\right)\right) \\
&\vee \left(f^k(\mathbf{x}) \le z^k \wedge f^k(\mathbf{y}) \le z^k \wedge f^k(\mathbf{w}) \le z^k\right) \\
&\Rightarrow \left(\left(f^k(\mathbf{x}) = f^k(\mathbf{w})\right) \vee \left(f^k(\mathbf{x}) \le z^k \wedge f^k(\mathbf{w}) \le z^k\right), \forall k \in [K]\right) \\
&\Rightarrow F(\mathbf{x}) =_{(Z)} F(\mathbf{w})
\end{aligned}
\tag{10}
$$

**Statement II:** $F(\mathbf{x}) \prec_{(Z)} F(\mathbf{y}) \wedge F(\mathbf{y}) \prec_{(Z)} F(\mathbf{w}) \Rightarrow F(\mathbf{x}) \prec_{(Z)} F(\mathbf{w})$

*Proof of Statement II.* According to Eq. (6), we have

$$
\begin{aligned}
&F(\mathbf{x}) \prec_{(Z)} F(\mathbf{y}) \Leftrightarrow \\
&\exists k_1 \in [K] : f^{k_1}(\mathbf{x}) < f^{k_1}(\mathbf{y}) \wedge f^{k_1}(\mathbf{y}) > z^{k_1} \wedge F^{k_1 - 1}(\mathbf{x}) =_{(Z)} F^{k_1 - 1}(\mathbf{y}) \\
&F(\mathbf{y}) \prec_{(Z)} F(\mathbf{w}) \Leftrightarrow \\
&\exists k_2 \in [K] : f^{k_2}(\mathbf{y}) < f^{k_2}(\mathbf{w}) \wedge f^{k_2}(\mathbf{w}) > z^{k_2} \wedge F^{k_2 - 1}(\mathbf{y}) =_{(Z)} F^{k_2 - 1}(\mathbf{w})
\end{aligned}
\tag{11}
$$

Let $k' = \min\{k_1, k_2\}$, according to Statement I and Eq. (11), we have:

$$
\left(F^{k'-1}(\mathbf{x}) =_{(Z)} F^{k'-1}(\mathbf{y})\right) \wedge \left(F^{k'-1}(\mathbf{y}) =_{(Z)} F^{k'-1}(\mathbf{w})\right) \Rightarrow F^{k'-1}(\mathbf{x}) =_{(Z)} F^{k'-1}(\mathbf{w})
\tag{12}
$$

According to Eq. (12), if $k' = k_1 < k_2$, we have

$$
\left(f^{k'}(\mathbf{x}) < f^{k'}(\mathbf{y}) = f^{k'}(\mathbf{w})\right) \wedge \left(z^{k'} < f^{k'}(\mathbf{y}) = f^{k'}(\mathbf{w})\right)
\tag{13}
$$

If $k' = k_2 < k_1$, we have

$$
\left(f^{k'}(\mathbf{x}) = f^{k'}(\mathbf{y}) < f^{k'}(\mathbf{w})\right) \wedge \left(z^{k'} < f^{k'}(\mathbf{w})\right)
\tag{14}
$$

Otherwise when $k' = k_1 = k_2$, we have

$$\left(f^{k'}(\mathbf{x}) < f^{k'}(\mathbf{y}) < f^{k'}(\mathbf{w})\right) \wedge \left(z^{k'} < f^{k'}(\mathbf{y}) < f^{k'}(\mathbf{w})\right) \tag{15}$$

Combing Eq. (12), Eq. (13), Eq. (14) and Eq. (15), we have found existing $k = k'$ such that $\left(f^k(\mathbf{x}) < f^k(\mathbf{w})\right) \wedge \left(f^k(\mathbf{w}) > z^k\right) \wedge \left(F^{k-1}(\mathbf{x}) =_{(Z)} F^{k-1}(\mathbf{w})\right)$.

**Statement III:** $F(\mathbf{x}) =_{(Z)} F(\mathbf{y}) \wedge F(\mathbf{y}) \prec_{(Z)} F(\mathbf{w}) \Rightarrow F(\mathbf{x}) \prec_{(Z)} F(\mathbf{w})$.

*Proof of Statement III.* According to the definitions of $=_{(Z)}$ and $\prec_{(Z)}$ in Eq. (5) and Eq. (6),

$$\begin{aligned}
&\left(F(\mathbf{x}) =_{(Z)} F(\mathbf{y})\right) \wedge \left(F(\mathbf{y}) \prec_{(Z)} F(\mathbf{w})\right) \Rightarrow \\
&\left(\forall k_1 \in [K] : \left(f^{k_1}(\mathbf{x}) = f^{k_1}(\mathbf{y})\right) \vee \left(f^{k_1}(\mathbf{x}) \le z^{k_1} \wedge f^{k_1}(\mathbf{y}) \le z^{k_1}\right)\right) \\
&\wedge \left(\exists k_2 \in [K] : f^{k_2}(\mathbf{y}) < f^{k_2}(\mathbf{w}) \wedge f^{k_2}(\mathbf{w}) > z^{k_2} \wedge F^{k_2-1}(\mathbf{y}) =_{(Z)} F^{k_2-1}(\mathbf{w})\right)
\end{aligned} \tag{16}$$

Then, we find a $k = k_2$, such that,

$$\begin{aligned}
&\left(\left(f^k(\mathbf{x}) = f^k(\mathbf{y}) < f^k(\mathbf{w})\right) \vee \left(f^k(\mathbf{x}) \le z^k < f^k(\mathbf{w})\right)\right) \\
&\wedge \left(z^k < f^k(\mathbf{w})\right) \wedge \left(F^{k-1}(\mathbf{x}) =_{(Z)} F^{k-1}(\mathbf{w})\right) \\
&\Rightarrow \left(f^k(\mathbf{x}) < f^k(\mathbf{w})\right) \wedge \left(z^k < f^k(\mathbf{w})\right) \wedge \left(F^{k-1}(\mathbf{x}) =_{(Z)} F^{k-1}(\mathbf{w})\right)
\end{aligned} \tag{17}$$

in which indicates $F(\mathbf{x}) \prec_{(Z)} F(\mathbf{w})$.

**Statement IV:** $F(\mathbf{x}) \prec_{(Z)} F(\mathbf{y}) \wedge F(\mathbf{y}) =_{(Z)} F(\mathbf{w}) \Rightarrow F(\mathbf{x}) \prec_{(Z)} F(\mathbf{w})$.

*Proof of Statement IV.* According to the definitions of $=_{(Z)}$ and $\prec_{(Z)}$ in Eq. (5) and Eq. (6),

$$\begin{aligned}
&\left(F(\mathbf{x}) \prec_{(Z)} F(\mathbf{y})\right) \wedge \left(F(\mathbf{y}) =_{(Z)} F(\mathbf{w})\right) \\
&\Rightarrow \left(\exists k_1 \in [K] : f^{k_1}(\mathbf{x}) < f^{k_1}(\mathbf{y}) \wedge f^{k_1}(\mathbf{y}) > z^{k_1} \wedge F^{k_1-1}(\mathbf{x}) =_{(Z)} F^{k_1-1}(\mathbf{y})\right) \wedge \\
&\left(\forall k_2 \in [K] : \left(f^{k_2}(\mathbf{y}) = f^{k_2}(\mathbf{w})\right) \vee \left(f^{k_2}(\mathbf{y}) \le z^{k_2} \wedge f^{k_2}(\mathbf{w}) \le z^{k_2}\right)\right)
\end{aligned} \tag{18}$$

Then, we can find a $k = k_1$, such that,

$$\begin{aligned}
&\left(f^k(\mathbf{x}) < f^k(\mathbf{y}) = f^k(\mathbf{w})\right) \wedge \left(z^k < f^k(\mathbf{w}) = f^k(\mathbf{y})\right) \wedge \left(F^{k-1}(\mathbf{x}) =_{(Z)} F^{k-1}(\mathbf{w})\right) \\
&\Rightarrow \left(f^k(\mathbf{x}) < f^k(\mathbf{w})\right) \wedge \left(z^k < f^k(\mathbf{w})\right) \wedge \left(F^{k-1}(\mathbf{x}) =_{(Z)} F^{k-1}(\mathbf{w})\right) \Rightarrow F(\mathbf{x}) \prec_{(Z)} F(\mathbf{w})
\end{aligned} \tag{19}$$

By substituting the conclusions in statement **(I)**, **(II)**, **(III)** and **(IV)** into (i), (ii), (iii), and (iv) in Eq. (9) respectively, we get:

$$\begin{aligned}
F(\mathbf{x}) \preceq_{(Z)} F(\mathbf{y}) \wedge F(\mathbf{y}) \preceq_{(Z)} F(\mathbf{w}) &\Rightarrow \left(F(\mathbf{x}) \prec_{(Z)} F(\mathbf{w})\right) \vee \left(F(\mathbf{x}) =_{(Z)} F(\mathbf{w})\right) \\
&\Rightarrow F(\mathbf{x}) \preceq_{(Z)} F(\mathbf{w})
\end{aligned} \tag{20}$$

which concludes the proof. $\square$

## B  Supplementary results

We provide the results for the feature selection task in bio-informatics (Section 4.1.3) under different random seeds in Figure 4.

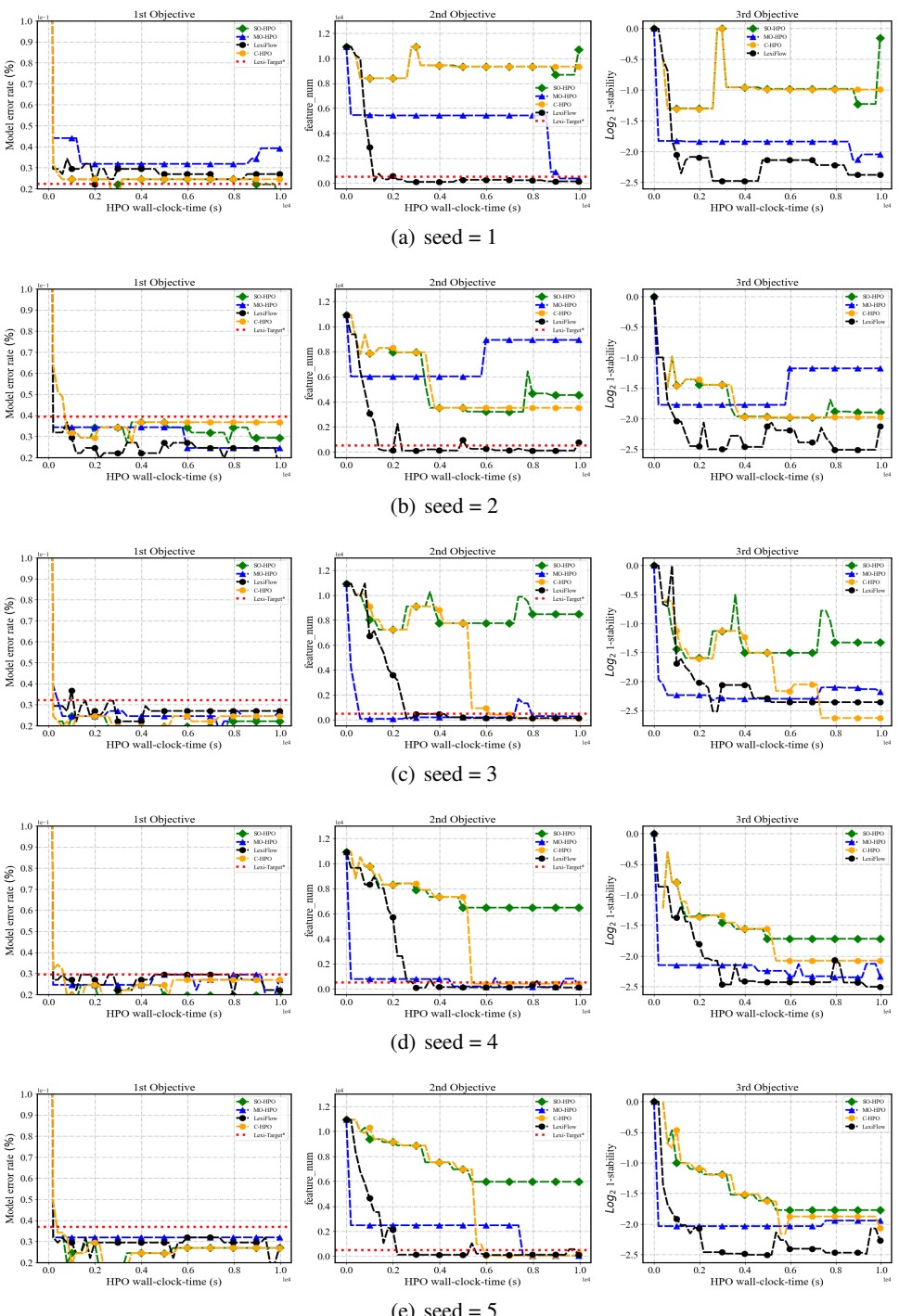

Figure 4: The detailed results in tuning Xgboost on biological dataset AP_Colon_Kidney with different random seeds.

## C    EXPERIMENTATION DETAILS

### C.1    SEARCH SPACE

The detailed search space in tuning Neural Networks, Random Forest, and XGboost are shown in Table 3, Table 4 and Table 5, respectively.

Table 3: Hyperparameters tuned in Neural Network

| hyperparameter | type | range |
|---|---|---|
| epoch num | int | [1, 20] |
| layer num | int | [1 , 3] |
| hidden units num per layer | int | [4, 128] |
| dropout value per layer | float | [0.2, 0.5] |
| learning rate per layer | float | [1e-5, 1e-1] |

Table 4: Hyperparameters tuned in Random Forest

| hyperparameter | type | range |
|---|---|---|
| max features | float | $[\min(0.1, 1/\sqrt{\text{data\_features}}), 1.0]$ |
| estimators number | int | [4, min(2048, train_datasize)] |
| max leaves | int | [4, train_size] |

Table 5: Hyperparameters tuned in XGboost

| hyperparameter | type | range |
|---|---|---|
| estimators number | int | [4, min(32768, train_datasize)] |
| max leaves | int | [4, min(32768, train_datasize)] |
| max depth | int | [0, 6, 12] |
| min child weight | float | [0.001, 128] |
| learning rate | float | [1/1024, 1.0] |
| subsample | float | [0.1, 1.0] |
| colsample by tree | float | [0.01, 1.0] |
| colsample by level | float | [0.01, 1.0] |
| reg alpha | float | [1/1024, 1024] |
| reg lambda | float | [1/1024, 1024] |

### C.2    DATE STATISTICS INFORMATION

All datasets used in our experiment are available in OpenML. In Table 6, we show the detailed statistics information of the datasets used in Section 4.2.

Table 6: Date statistics information

| Dataset statistics | Gisette | Christine | Scene | Ginal_prior | Ginal_agnostic | Bioresponse | Hiva_agnostic | Madelon |
|---|---|---|---|---|---|---|---|---|
| # of train instance | 4900 | 4063 | 1805 | 2601 | 2601 | 2813 | 3171 | 1950 |
| # of val instance | 2098 | 1355 | 602 | 867 | 867 | 938 | 1058 | 650 |
| # full feature dimension | 5000 | 1636 | 299 | 784 | 970 | 1776 | 1617 | 500 |

### C.3    DETAILS OF THE VALIDATION LOSS CALCULATION IN SECTION 4.1.2

In section 4.1.2, in order to keep the same experiment setting with the paper (Brookhouse & Freitas, 2022), we use the accuracy metric named *the geometric mean of Sensitivity and Specificity* from

(Brookhouse & Freitas, 2022) to calculate the 1st objective validation loss in the experiment section. More specifically, the accuracy metric *geometric mean of Sensitivity and Specificity* $GM_{Sen \times Spe}$ is calculated by:

$$\text{Sensitivity} = \frac{TP}{TP + FN}, \text{Specificity} = \frac{TN}{TN + FP} \tag{21}$$
$$GM_{Sen \times Spe} = \sqrt{\text{Sensitivity} \times \text{Specificity}}$$

Where $TP$, $TN$, $FN$ and $FP$ represent the number of true positive instances, the number of true negative instances, the number of false negative instances and the number of false positive instances, respectively. The validaiton loss reported in Table 1 is calculated by $1 - GM_{Sen \times Spe}$. More information of this metric could be found in (Brookhouse & Freitas, 2022).

