# OpenReview forum: "Targeted Hyperparameter Optimization with Lexicographic Preferences Over Multiple Objectives"
_ICLR.cc/2023/Conference — ICLR 2023 notable top 5%_

### Official Review · Reviewer_NjRw · 2022-10-18

**Confidence:** 4
**Correctness:** 3
**Technical Novelty And Significance:** 3
**Empirical Novelty And Significance:** 3
**Recommendation:** 8

**Clarity, Quality, Novelty And Reproducibility:**

The paper is written to a high standard and is easy to understand. In terms of originality, I think the LexiHPO setup is an interesting new take on multi-objective optimization. The algorithm presented is perhaps less original, seeming to be mostly an adaptation of recent work in randomized direct search to this new setup. In terms of reproducibility, I have checked the anonymous repository associated with the paper and it seems that code is provided to reproduce all results and baselines.

**Strength And Weaknesses:**

Strengths:
- The LexiHPO framework is novel to the best of my knowledge, and may be more intuitive to practitioners than dealing with multi-objective approaches that discover Pareto frontiers.
- The LexiHPO algorithm is described well and the authors provide intuition about how it was designed.
- Extensive experimental results are provided that demonstrate the performance of the algorithm, but also the use-cases presented help to justify the idea of introduce a lexicographic ordering into multi-objective optimization.

Weaknesses:
- Only a few sentences are devoted to existing work regarding multi-objective optimization with preferences. I think the paper would be improved if a little more spaced was devoted to explaining why LexiHPO is somehow a better way to incorporate preferences than Paria et al. 2020, Abdolshah et al. 2019 and Zitzler et al. 2008, and also why none of these methods could be used as experimental baselines.
- In Section 3 it is stated: "However one problem with the vanilla lexicographic relations is that they cannot accommodate user-specified tolerances and goals on the objectives.". I found the flow here a little confusing - wouldn't it be better to introduce equations (5), (6), (7) in Section 2.1 since one can argue they fundamentally relate to the setup itself, rather than the algorithm. Whereas the algorithm is responsible for computing the online statistics in equation (8)?
- The datasets used in Section 4.2 are very small and the results are not entirely convincing statistically. While I feel that the contributions of the paper stand without this section, I think that in order to convincingly argue that LexiFlow can be used to reduce overfitting it would be necessary to use more datasets and/or perform rigorous statistical hypothesis testing.

Minor corrections:
- In the example at the bottom of page 3 it is stated the primary objective is the model's prediction error, but later in the example it is referred to as "loss".
- In the same example, it is stated that the goal vector should contain +infinity in the first position, shouldn't this be -infinity?
- "We also urge methods that are efficient and cost-frugal" -- "urge" is probably the wrong choice of word in this sentence.
- In Figure 2 and Figure 3 - is there a reason why the LexiTarget is not shown for all objective?
- Section 4.1.3 last paragraph - Figure 4 is referenced whereas I think it should be Figure 3?


**Summary Of The Paper:**

This paper introduces a new formulation of hyperparameter optimization over multi-objectives, LexiHPO, in which the user can specify a lexicographic ordering regarding the priority of each objective. The paper then introduces an algorithm for solving this problem, LexiFlow, which seems to be an extension of recent work on randomized directed search (Wu et al. 2021) to account for the new lexicographic setup. This algorithm is then evaluated on a number of different multi-objective HPO tasks in which the objectives are ordered: finding accurate and fast/small neural networks, finding accurate and fair gradient boosting machines, finding accurate models on a bioinformatics tasks that use a small number of features. Additionally, the paper argues that LexiFlow can be used as a tool to reduce overfitting by transforming a single-objective HPO tasks (e.g. minimizing validation loss) into a dual-objective optimization task in which the second objective is designed to reduce overfitting to the validation set (e.g., by restricting the number of features). Experimental results are provided to demonstrate this idea.

**Summary Of The Review:**

This is a well-written paper with an interesting new perspective on multi-objective hyperparameter optimization. The paper could be improved with more discussion of how their setup relates to the existing work of multi-objective optimization with preferences, and the experimental results in Section 4.2 could be made more convincing by using more datasets and/or statistical hypothesis testing.

---

> ### Author Response · Authors · 2022-11-14
> **Response to AnonReviewer NjRw**
>
> We thank the reviewer for the constructive comments. We have made revision to the paper accordingly. In addition, please find our response to some of the overall comments below.
>
> **[Re Weakness 1: Only a few sentences are devoted to existing work regarding multi-objective optimization with preferences]**
>
> Thanks for your suggestions. We have revised related works in our manuscript according to the suggestions.
> We did not include them as our baselines because of the following two reasons:
>
> (1) The preferences defined in Paria et al. 2020, Abdolshah et al. 2019, and Zitzler et al. 2008 are not lexicographic preferences and thus these works can not directly apply to the Lexi-HPO problem studied in this work.
>
> (2) The source code for Abdolshah et al. 2019 and Zitzler et al. 2008 are not publically available. Paria et al. 2020 provide publicly available source code, but this work requires a probability distribution over the set of Pareto optimal values to encode user preference on the objectives. In practice, how to specify such a probability distribution to solve the Lexi-HPO problem is unclear. This prevents us from making a comparison between Paria et al. 2020 and our method. We thus further omit the comparison with Paria et al. 2020.
>
> **[Re Weakness 2: Wouldn't it be better to introduce equations (5), (6), (7) in Section 2.1.]**
>
> We thank the reviewer for the suggestion. What the reviewer suggested is indeed an alternative way to organize the writing. However, Eq (5)(6)(7) is actually not necessary from the problem description perspective. We choose to only introduce the essential definitions and operations needed for characterizing the problem itself in Section 2.1. Eq (5)(6)(7) are newly proposed relations satisfying some non-trivial properties and we use Eq (5)(6)(7) as a part of our algorithm.
>
> **[Re Weakness 3: Small datasets in Section 4.2 ]**
>
> We appreciate the reviewer's constructive comments. We would like to first highlight that although the number of data instances for the evaluated datasets is small, the feature numbers of the datasets are quite large (with an average of 1601 across the 9 datasets). In addition, overfitting is more likely to happen on datasets with small data samples (Xue, 2019). We are actively seeking collaboration with ML practitioners to perform more real practical evaluations in the future.
>
> **[Re Minor Corrections 4: Why is the LexiTarget not shown for all objectives in Figure 2 and Figure 3? ]**
>
> We do not show Lexi-target for all objectives in Fig.2 and Fig.3 because we do not provide goals and tolerances for some objectives (2nd objective in Fig.2 and 3rd objective in Fig.3). We expect to minimize these objectives without any compromise if the high-priority objective is well optimized (reaching corresponding Lexi-targets).
>
> **[Re Other Minor Corrections]**
>
> We thank the reviewer for carefully reading our paper and pointing out the wording, notation, and consistency issues. We have made revisions to the manuscript accordingly
>
> *Ying, Xue. "An overview of overfitting and its solutions." In Journal of Physics: Conference series, vol. 1168, no. 2, p. 022022. IOP Publishing, 2019.*

---

> > ### Comment · Reviewer_NjRw · 2022-12-06
> > **Thank you for the response**
> >
> > Thank you for the clarifying comments and references. It gives me further confidence to recommend that the paper is accepted.

---

### Official Review · Reviewer_Fdr2 · 2022-10-21

**Confidence:** 4
**Correctness:** 3
**Technical Novelty And Significance:** 4
**Empirical Novelty And Significance:** 4
**Recommendation:** 8

**Clarity, Quality, Novelty And Reproducibility:**

Clarity:

Clarity is mixed.  The introduction and formulation of this problem are mostly clear. However the description of the algorithm and, to a lesser extent, the experiments is rather difficult to parse (see comments above).


Quality/Novelty:

The problem is important and the solution appears novel, and while the paper has some issues I would say that the quality is good overall.

Reproducibility:

Given the details in the paper I am reasonably certain that I could reproduce at least a reasonable representation of the results.

**Strength And Weaknesses:**

Strengths:

- The problem is well motivated and thoroughly described; and the proposed use of lexicographic preferences is well justified.
- The problem formulation is mostly clear (but see caveats below).
- In general I was able to follow the development of the algorithm and its description.


Weaknesses:

- The algorithm and its description are somewhat hard to parse.  After multiple read-throughs I think I have a reasonable understanding (but not perfect - see questions below), but it would have been easier if the larger "text chunks" (e.g. paragraph 2 on page 5) were rewritten using dot-points, tables etc to improve readability.  You may also want to consider motivating the approach taken in the algorithm (what is it based on and why) before presenting it.
- Have you considered the convergence properties of this algorithm ("big-O" behaviour, regret bounds etc).


Minor points:

- The presentation was inconsistent in the results section - for example the highlighting in table 1 (bold for all within the error bars of the best result) and table 2 (bold only for the "best", ignoring error bars).
- Why inf (text) in some places, $\infty$ in others?


Questions:

- Regarding (1), do you intentionally allow for solutions that do not reach the goal for some objectives?  This appears to be the case (z_*^k is max of the goal and "best we can do plus tolerance" if I'm reading it correctly) - perhaps some commentary on why such solutions are acceptable would be of assistance here?
- Was there a reason (technical or otherwise) for using randomised direct search rather than an algorithm specifically designed to deal with expensive optimisation problems (Bayesian optimization or similar)?
- You mention that the solution found is (potentially) simply one sample of a region of the Pareto front.  What if the algorithms finds multiple such solutions?  Or is this not possible?
- What is the purpose of lines 9-12 in the algorithm?  In particular, where do the various "update equations" here come from?
- What is the termination condition for the algorithm?

**Summary Of The Paper:**

In machine learning it is relatively common to have multiple objectives, typically arranged in some loose descending order of importance - for example we typically care most about accuracy, but feature counts, fairness, robustness etc are also important.  Multiobjective optimisation is the obvious way to approach such problems, but can be wasteful as it will explore the entire Pareto front, not just the regions the operator is interested in, so for example it will spend just as much time exploring extremely fair solutions with terrible accuracy - or vice-versa - as it will spend exploring more balanced solutions.

The paper proposes using lexicographic preferences as a natural way to express the ordering over the objectives, and subsequently proposes an algorithm that seeks solutions satisfying these lexicographic preferences rather than just a batch of Pareto optimal "solutions".

**Summary Of The Review:**

Overall I think this is a good paper on an important problem.  I have some reservations regarding the clarity of the presentation of the algorithm, but on balance the positives outweigh the negatives.

---

> ### Author Response · Authors · 2022-11-14
> **Response to AnonReviewer Fdr2 - Part 1**
>
> We thank the reviewer for the insightful comments. We have revised the paper accordingly. In addition, please find our response to some of your comments below.
>
> **[Re Weakness 1: Clarity of algorithm description]**
>
> Thanks for your suggestions. We have revised the manuscript according to your suggestions as summarized in the overall response.
>
> **[Re Weakness 2: Convergence properties of the algorithm ("big-O" behavior, regret bounds etc)]**
>
> Thanks for your suggestions. We indeed considered its convergence properties, and we think it is meaningful but non-trivial. We would like to share some initial thoughts on the convergence analysis of our method: Due to the existence of multiple objectives and multiple levels of "optimality", we need to be creative in the definition of convergence. For example, we could define different levels of convergence, such as how fast we approach $X_*^1$, $X_*^2$, .... With such a definition, we can characterize the convergence rate to $X_*^{i+1}$ after we reach $X_*^{i}$.  (convergence rate is $O(\frac{\sqrt{d}}{\sqrt{K}})$ where $K$ is the iteration number and $d$ is the dimension of the hyperparameter space for approaching $X_*^{1}$ according to the convergence conclusion in the single objective case). We consider a rigorous analysis of the convergence property of our method as an interesting future follow-up work.
>
> **[Re Question 1: Do you intentionally allow for solutions that do not reach the goal for some objectives? Some commentary on why such solutions are acceptable would be of assistance here]**
>
> Thank you for your thoughtful question and constructive suggestion.  We have made it clearer by adding a footnote in Eq (1) according to your suggestion.
>
> **[Re Question 2: Was there a reason (technical or otherwise) for using randomized direct search rather than an algorithm specifically designed to deal with expensive optimization problems (Bayesian optimization or similar)?]**
>
> There are two considerations when designing the solution:
>
> (1) It is proven impossible to use a utility function to fully represent the lexicographic optimization problem (Shi et al. 2020) in general. And a utility function is a basis for many classical HPO algorithms like bayesian optimization.
>
> (2) Considering (1), we further find comparisons between any two feasible hyperparameter configurations are always conclusive in the Lexi-HPO problem.
>
> We take advantage of this unique property in our proposed randomized direct search method.
> Randomized direct search based algorithms could overcome the difficulty as illustrated in (1)  and leverage the unique property in (2).
>
> *Shi, Bowen, Gaowang Wang, and Zhixiang Zhang. "On the Utility Function Representability of Lexicographic Preferences." (2020).*

---

> > ### Author Response · Authors · 2022-11-14
> > **Response to AnonReviewer Fdr2 - Part 2**
> >
> > **[Re Question 3: You mention that the solution found is (potentially) simply one sample of a region of the Pareto front. What if the algorithm finds multiple such solutions? Or is this not possible?]**
> >
> > Thank you for your comments. There may exist multiple lexi-optimal solutions according to our definition. The multiple lexi-optimal solutions have exactly the same performance on all the objectives. One shall be satisfied as long as one of them is found. Based on this understanding, our method thus only tries to output one lexi-optimal solution. It is easy to modify the algorithm to output all such solutions it found instead of one if needed.
> >
> > **[Re Question 4 and 5: What is the purpose of lines 9-12 in the algorithm? In particular, where do the various "update equations" here come from? What is the termination condition for the algorithm?]**
> >
> > Line 9-12 are for adjusting step size and random restart following existing work [1][2][3]. The algorithm will terminate when it runs out of resource budget. We explained these in paragraph 2 on page 5 in the original submission:
> > “Following the same spirit of an existing randomized direct search based single-objective HPO method (Wu et al., 2021), LexiFlow includes the restart and dynamic step size techniques to free the algorithm from local optimum and manual configuration of stepsize (line 9-11 of Alg. 1).” “The algorithm terminates if it runs out of resource budget and outputs a Lexi-optimal configuration $x^*$, which is maintained in a way such that it is Lexi-optimal among all the evaluated configurations.” We have reorganized this large text chunk into bullet points to make the information more obvious.
> >
> > **[Re Minor points 1:  the highlighting in table 1 (bold for all within the error bars of the best result) and table 2 (bold only for the "best", ignoring error bars) is inconsistent]**
> >
> > We compare the values in table 1 and table 2 in different ways and thus bold them using different standards. In table 1, we expect to compare two optimizing objectives with lexicographic preference across different methods, so we bold for all within the error bars of the best result in 1st objective. In table 2, we intend to compare the final test accuracy after optimizing the validation loss (1st objective) and feature number (2nd objective), so we bold the results with the best test accuracy which is not part of the lexicographic objectives and does not have a tolerance threshold input.
> >
> > *[1] Martí, R., J. Marcos Moreno-Vega, and A. Duarte. "Advanced multi-start methods." Handbook of metaheuristics. Springer, Boston, MA, 2010*
> >
> > *[2] György, András, and Levente Kocsis. "Efficient multi-start strategies for local search algorithms." Journal of Artificial Intelligence Research 41 (2011): 407-444.*
> >
> > *[3] Wu, Qingyun, Chi Wang, and Silu Huang. "Frugal optimization for cost-related hyperparameters." Proceedings of the AAAI Conference on Artificial Intelligence. Vol. 35. No. 12. 2021.*

---

> > > ### Comment · Reviewer_Fdr2 · 2022-11-17
> > > **Response to author**
> > >
> > > Thank you for the thoughtful response.  I feel that the changes made to the paper have made it significantly easier to read - in particular having read the new version I feel I have a much better understanding of the algorithm.

---

### Official Review · Reviewer_GSAk · 2022-10-25

**Confidence:** 4
**Correctness:** 4
**Technical Novelty And Significance:** 4
**Empirical Novelty And Significance:** 4
**Recommendation:** 8

**Clarity, Quality, Novelty And Reproducibility:**

The paper is overall well structured and well written. The explanations are easy to follow and the logic flows are reasonable.

The quality of the paper is very good. The method is well motivated, however, still imposing some assumptions which might be non-trivial. Still, for such a first work, such strong assumptions might be reasonable and be resolved in follow-up work.

The methodology is novel to the best of my knowledge.

A good level of detail is given in the paper such that the work should be reproducible.

**Strength And Weaknesses:**

Strengths:
- Interesting method with a lot of potential
- Strong performance compared to existing state-of-the-art multi-objective HPO methods
- Robust and strong anytime performance

Weaknesses:
- Requires the order of objective functions as an input by the user (who might not be able to express such an order)
- Requires the user to specify goals/thresholds which might be difficult to provide for black-box objectives. Meaning to provide such information the user needs extensive knowledge about the black-box objectives.

**Summary Of The Paper:**

The paper "Targeted Hyperparameter Optimization with Lexicographic Preference Over Multiple Objectives" proposes a novel algorithm for optimizing hyperparameters with multiple objective functions. In their approach, the authors assume a total order over the objective functions to be given and accordingly sort candidate hyperparameter configurations in lexicographic order. To optimize the objectives effectively the authors propose a directed search algorithm and find their method LexiFlow to produce strong results.

**Summary Of The Review:**

All in all, I think that this paper makes an interesting contribution and the proposed method shows reasonable performance across tuning tasks and also within tuning tasks compared to state-of-the-art methods.

---

> ### Author Response · Authors · 2022-11-14
> **Response to AnonReviewer GSAk**
>
> We thank the reviewer for highlighting the appealing properties of our method. We also partially agree with the reviewer on the weaknesses of our work but would like to elaborate a little bit more on them.
>
> **[Re Weakness 1:  Requires the order of objective functions as an input by the user (who might not be able to express such an order]**
>
> LexiFlow covers a wide range of scenarios that fall into the following two categories: (1) To directly perform lexicographic optimization in cases where the users do have lexicographic preferences over multiple objectives. (2) To optimize a particular objective that is inaccessible during the optimization, by finding lexi-optimal solutions in terms of a list of proxy objectives. (as mentioned in the experiments section). These two categories of use cases cover a very wide range of scenarios of practical interest. But we do agree with your point that there do exist cases where users cannot characterize their needs through the priority orders on the objectives. We consider those cases as interesting future work.
>
> **[Re Weakness 2: Requires the user to specify goals/thresholds which might be difficult to provide for black-box objectives]**
>
> Goals and thresholds are optional and we allow users to omit one or both as mentioned in Remark 2. We agree that it requires some understanding of the objectives to be able to specify goals/thresholds. The required understanding is about what objective values are good enough, what difference matters etc. It doesn't require knowing the relationship between the objective values and the hyperparameters which is a blackbox.

---

### Author Response · Authors · 2022-11-14
**Overall Response - Paper Revision Summary**

We thank the reviewers for the encouraging and constructive comments. We made the following revisions to the manuscript based on the reviewers' comments:

Based on reviewer Fdr2’s suggestions,
1. We improved the clarity of the algorithm description in both pseudo-code in Algorithm 1 and the descriptive text. (cf reviewer Fdr2’s comment on the first weakness and the last question).
2. We improved notation consistency. (cf reviewer Fdr2’s comment on the minor points).
3. We added a footnote on the definition of lexi-trarget value $z^k_*$ in section 2.1. (cf reviewer Fdr2’s first question).

Based on reviewer NjRw’s comments,
1. We added more sentences to introduce several relevant existing work in section 1.1 (cf reviewer NjRw’s comments on the first weakness).
2. We fixed several wording, notation and consistency problems. (cf reviewer NjRw’s comments in the minor corrections).

---

### Decision · Program_Chairs · 2023-01-20

**Decision:**

Accept: notable-top-5%

**Justification For Why Not Higher Score:**

N/A

**Justification For Why Not Lower Score:**

* Unanimously supported by the reviewers, and confidently so.
* Strong performance gains in an important and widespread problem (multi-objective hyperparameter optimization).

**Metareview: Summary, Strengths And Weaknesses:**

The reviewers and meta reviewer all appreciated the quality of the work with its strong empirical results in a clear, well-written manuscript. The proposed multi-objective HPO method will inform future research. They thank the authors for their response and their efforts during the rebuttal phase, which further improves the submission. They enthusiastically recommend the paper for acceptance.

**Note From Pc:**

if the above contains the word "oral" or "spotlight" please see: "oral" presentation means -> notable-top-5% and "spotlight" means -> notable-top-25%. As stated in our emails, we are disassociating presentation type from AC recommendations